# Offline Multi-Agent Reinforcement Learning with Knowledge Distillation

**Wei-Cheng Tseng[1], Tsun-Hsuan Wang[2], Lin Yen-Chen[2], Phillip Isola[2]**
[1]University of Toronto, [2]MIT CSAIL
weicheng.tseng@mail.utoronto.ca, {tsunw,yenchenl,phillipi}@mit.edu

## Abstract

We introduce an offline multi-agent reinforcement learning (offline MARL) framework that utilizes previously collected data without additional online data collection. Our method reformulates offline MARL as a sequence modeling problem and thus builds on top of the simplicity and scalability of the Transformer architecture. In the fashion of centralized training and decentralized execution, we propose to first train a teacher policy who has the privilege to access every agent's observations, actions, and rewards. After the teacher policy has identified and recombined the "good" behavior in the dataset, we create separate student policies and distill not only the teacher policy's features but also its structural relations among different agents' features to student policies. We show that our framework significantly improves performances on a range of tasks and outperforms state-of-the-art offline MARL baselines. Furthermore, we demonstrate that the proposed method has a better convergence rate, is more sample efficient, and is more robust to various demonstration qualities compared with baselines.

## 1 Introduction

The *online* learning paradigm assumed by existing multi-agent reinforcement learning (MARL) algorithms is one of the biggest obstacles to their widespread adoption. To apply MARL, one has to repeatedly perform the following two steps: (a) collect experiences by deploying multiple agents, typically with their latest learned policies, and (b) use the collected experiences to improve the policies. In many scenarios, frequently performing the first step is impractical because deploying multi-agents to the environment can be expensive and dangerous (e.g., self-driving cars). Therefore, devising *offline* MARL algorithms that can simply learn from previously collected datasets without interaction with the environment is an important step toward solving real-world problems.

Recently, several works have extended offline RL algorithms under the single-agent setting to offline MARL [45]. These works focus on addressing the distribution shift issue that causes the values of unseen state-action pairs to be erroneously estimated. However, these methods are based on temporal difference (TD) learning and thus require bootstrapping for credit assignment, a problem that is especially challenging under the multi-agent setting as the interactions between the agents and the environment can be highly complex.

To mitigate these issues, we explore the possibility to transform offline MARL into a sequence modeling problem. This paradigm shift, first introduced by Decision Transformer [3] under the single-agent setting, allows us to bypass bootstrapping and perform credit assignment directly via self-attention. A concurrent work, MADT [26], proposes to adapt Decision Transformer to the multi-agent setting by (a) sharing model parameters across agents and (b) attaching one-hot agent IDs to observations. Different from MADT, we propose a framework based on policy distillation that achieves better performance (as demonstrated in the experiments).

Our method first trains a teacher policy (instantiated as a decision transformer) to model the entire offline MARL dataset sequentially by accessing each agent's observation, action, and reward. The

36th Conference on Neural Information Processing Systems (NeurIPS 2022).

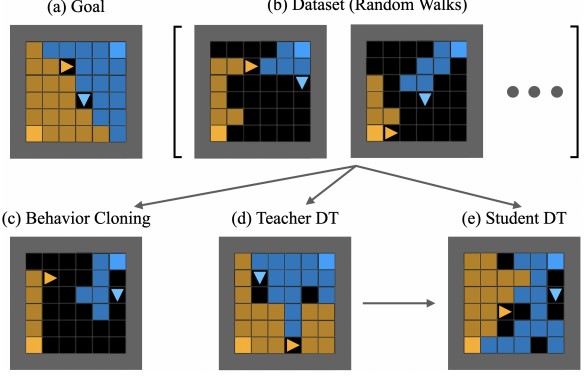

(a) Goal  (b) Dataset (Random Walks)

(c) Behavior Cloning  (d) Teacher DT  (e) Student DT

Figure 1: **Illustrative Example.** (a) Two agents are tasked to explore as many blocks as possible. Explored blocks are colored (e.g., blue). (b) The training dataset consists of agents' random walk trajectories and per-step rewards. (c) Agents trained with behavior cloning result in suboptimal performance. (d) A centralized Decision Transformer (Teacher DT) achieves superior performance but assumes privileged information. (e) Our framework structurally distills Teacher DT's policy into student policies for decentralized execution.

privilege to access every agent's information helps the teacher policy understand the underlying interaction across agents and predict actions that encourage cooperative behavior. However, the teacher policy cannot be deployed at test time in the fashion of decentralized execution. To address this issue, we initialize a separate student policy for each agent and distill the knowledge of the teacher policy into the student policy. In addition to allowing decentralized execution, this step is helpful because (a) the teacher policy can identify and recombine "good" behavior in a suboptimal dataset [3], (b) the centralized teacher allows credit assignment across agents via self-attention, and (c) distillation from a privileged model (teacher) provides richer and more stable learning signal to the student policy [2]. Furthermore, we propose a novel distillation objective that transfers structural relations of student policies' features rather than actual values of individual features. In our empirical results, we show that this objective is complementary to the classical policy distillation and helps us outperform state-of-the-art baselines on a range of benchmarks. Additionally, we provide further analysis on convergence rate, sample efficiency, and robustness to different demonstration qualities. Qualitative results are presented in our project website[1].

In summary, our contributions are as follows:

- A framework that reformulates offline MARL as sequential modeling and policy distillation for using self-attention to perform multi-agent credit assignment.

- A novel multi-agent policy distillation objective that focuses on preserving structural relationships among policies.

- State-of-the-art results on a broad range of tasks in terms of performance, convergence rate, sample efficiency and robustness to demonstration quality compared with baselines.

## 2 Related Works

**MARL.** Applying RL to multi-agent settings has been an active research domain for years. One line of work adapts value-based RL algorithms to perform credit assignment among agents given the joint reward signal [40, 9, 32, 38, 13]. COPA [23] and REFIL [15] uses dynamic team composition to further improve the training efficiency for tasks that require cooperative behaviors.

Another line of work extends the actor-critic framework to the multi-agent setting. For instance, MADDPG[25] uses DDPG [21] to jointly update decentralized policies and a centralized critic. For better scalability, several works [16, 17] further introduce the attention mechanism into the critic to prevent noises from irrelevant information .

A unique challenge in multi-agent environments [14, 30] is local observation. Previous works have proposed various communication mechanisms to tackle this issue. By exchanging information between agents, they can better understand the tasks and other agents. Some works [17, 8, 28, 24, 20] incorporate the attention mechanism to improve communication efficiency, while others [37] design a gating mechanism to decide when to communicate.

**Offline RL.** Different from the typical RL setting, offline RL trains the policy with a static, collected dataset and does not need any online interaction with the environment. Then, this policy is leveraged

---

[1]`https://weichengtseng.github.io/project_website/neurips22/index.html`

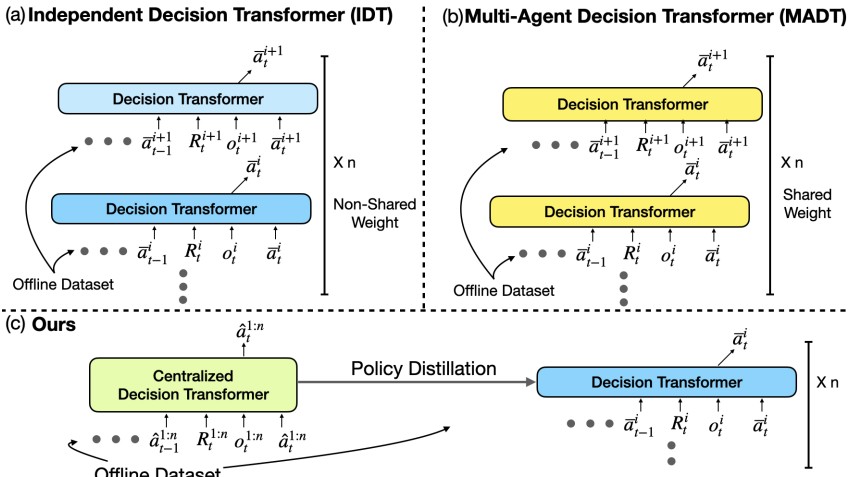

Figure 2: **Overview.** We compare (a) independent decision transformer (IDT), (b) multi-agent decision transformer [26] (MADT), and (c) our approach. (a) IDT trains an independent decision transformer for each agent separately. (b) MADT extends IDT by sharing parameters across multiple agents and concatenate agent's one-shot IDs to the observations. (c) Our approach first train a teacher policy, instantiated by a centralized decision transformer, and then distill both its features and structural relations among features to IDT.

to interact with the online environment to obtain promising results or assist exploration [43]. However, trajectories existing in offline datasets and interaction with the online environment have different distributions. One of the strategies to mitigate this issue is applying policy constraints [27, 48]. Another direction to alleviate this difficulty is to regularize the value estimation in reinforcement learning [18] or take uncertainty into consideration [1].

Recently, the Decision Transformer [3] outperforms many state-of-the-art offline RL algorithms by regarding the training process as a sequential modeling phase and testing on the online environment. This approach can bypass the drawback of TD-learning and overcome sparse reward difficulty. MADT [26] further extends Decision Transformer to a multi-agent domain by making agents controlled by a shared weight transformer-based policy.

**Knowledge Distillation.** Knowledge distillation (KD) transfers knowledge from one deep learning model (the teacher) to another (the student). The objective originally proposed by [12] minimizes the KL divergence between the teacher and student outputs. This formulation makes intuitive sense when the output is a distribution, such as a probability mass function over classes. Recent works focus on how to leverage the relationships between instances in the dataset to further provide meaningful representation learning [42, 41]. RKD [29] extracts the relation between instances from the feature space by defining distance-wise relation and angle-wise relation.

Knowledge distillation is also adapted to RL scenario. [6] proposes to distill multiple task-specific policies into a single policy which is more parameter-efficient. DPD [19] utilizes two policies that interact with the same environment with different initialization to explore different perspectives of the environment and extract knowledge from each other to enhance their learning. M&M [7] combines curriculum learning as well as distillation to allow agents to perform well in an environment with large action space.

The typical goal of knowledge distillation is model compression by making the student model smaller. Our method is different because our goal is to enable decentralized execution in the multi-agent setting.

## 3 Method

In this section, we introduce our framework based on sequence modeling and policy distillation. We first describe a baseline, Independent Decision Transformer (IDT), that naively adapts Decision Transformer to the multi-agent setting in Section 3.1. Then, we present our novel *structural relation*

**Algorithm 1** Our Offline MARL

---

**Input:** offline dataset $D : \{\tau_i(o_t^i, a_t^i, r_t^i)_{t=1}^T\}_{i=1}^n$
**Initialize:** $\theta$ as the parameters of $\pi_{\text{teacher}}$ , $\phi^{1:n}$ as the parameters of $\pi^{1:n}$

// training centralized decision transformer
**for** $\tau$ in $D$ **do**
    $\hat{a}_t^{1:n} = \text{argmax} \, \pi_{centralized}(a^{1:n}|\hat{a}_{<t}^{1:n}, \theta)$
    $\theta = \theta - \alpha \, \nabla_\theta L_{\text{centralized}}$
**end for**

// training decision transformers for agents
Freeze the weight $\theta$
**for** $\tau$ in $D$ **do**
    $\hat{a}_t^{1:n} = \text{argmax} \, \pi_{\text{teacher}}(a^{1:n}|\hat{a}_{<t}^{1:n}, \theta)$
    **for** i=1:n **do**
        $\overline{a}_t^i = \text{argmax} \, \pi_i(a^i|\hat{a}_{<t}^i, \phi^i)$
        $\phi^i = \phi^i - \alpha \, \nabla_{\phi^i}(L_{action}^i + \alpha L_{rel}^i + \beta L_{KL}^i)$
    **end for**
**end for**

---

*distillation* for multi-agent policy distillation in Section 3.2. The overall algorithm is presented in Algorithm 1.

### 3.1  Independent Decision Transformer

An intuitive method to transform offline MARL into a sequence modeling problem is to treat each agent's trajectory as independent sequence. For each agent $i$, a separate decision transformer $\pi^i$ is trained to predict the next action $\overline{a}_{t+1}^i$ given the past trajectories $(o_{(t-B):t}^i, \overline{a}_{(t-B):t}^i, r_{(t-B):t}^i)$ where $B$ stands for the maximum number of past steps to consider. Since these agents are trained independently without access to other agents' information, we call this method Independent Decision

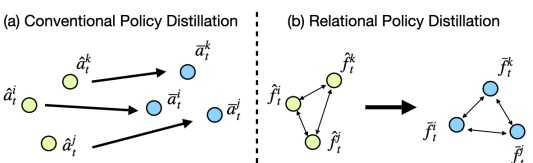

Figure 3: **Policy distillation objectives** we consider in our work. (a) Convential policy distillation [34] transfers individual outputs from a teacher model (green dots) to a student model (blue dots) point-wise, while (b) the proposed Relational Policy Distillation transfers structural relations of multi-agents' features.

Transformer (IDT). Its learning objective can be formulated as a sequence modelling problem that aims to predict the next action, $L_{action}^i = ||\overline{a}_{t+1}^i - a_{t+1}^i||_2$ We note that although each agent can leverage self-attention to perform credit assignment along their own trajectories, this algorithm cannot perform credit assignment *across agents*. This limitation prohibits agents to achieve cooperative behavior that attains maximal returns.

### 3.2  Teacher-Student Policy Distillation

To address IDT's cross-agents credit assignment issue, we propose to first train a centralized teacher policy $\pi_{\text{teacher}}$ that takes in all agents' observations, actions, and rewards $(o_{(t-B):t}^{1:n}, a_{(t-B):t}^{1:n}, r_{(t-B):t}^{1:n})$ combined via concatenation to predict all agents' actions $\hat{a}_{t+1}^{1:n}$ at the next step. We represent the teacher policy $\pi_{\text{teacher}}$ with a decision transformer and train it to minimize the prediction error of all agents' actions. Since the teacher policy has access to all agents' information, it can perform credit assignment *across agents* with self-attention to better foster cooperative behavior. This is similar to how centralized critic helps actor critic-based methods in online MARL [16, 25]. However, the centralized teacher policy cannot be deployed distributedly. To fix this, we propose to distill the teacher policy's features to $n$ separate student policies. We note that even though student policies operate independently during test time, they are very different from policies learned by IDT because (a) the supervision comes from a teacher policy that can perform credit assignment across agents, and (b) distillation provides a more stable learning signal to the student policy [2]. One intuitive policy distillation objective is to minimize the KL divergence between the actions predicted by the teacher

policy $\hat{a}^i$ and the student's policy $\overline{a}^i$:

$$L^i_{KL} = D_{KL}(\hat{a}^i | \overline{a}^i) \tag{1}$$

The other common distillation objective is to minimize the euclidean distance between the teacher's features $\hat{f}^i$ and the student's features $\overline{f}^i$ [33]:

$$L^i_{\text{feature}} = \|\hat{f}^i - \overline{f}^i\|^2_2 \tag{2}$$

However, both of these widely-used distillation strategies do not consider the structural relation between multi-agent students.

**Structural Relation Distillation.** To help the student policies learn as much as possible from the teacher policy, we introduce a new distillation objective tailored to the multi-agent setting. Our main insight is that in addition to transferring the teacher policy's actual feature values, we wish to preserve the structural relation among multi-agents' features. For example, if two agents belong to the same type of units and have similar observations in SMAC [35], their outputs should be close to each other. To this end, we define the relation between agents as the angle between feature vectors, and we intend to make the relation between student policies $\pi_{1:n}$ mimic the relation between agents controlled by $\pi_{\text{teacher}}$. In other words, we would like to minimize

$$L^i_{rel} = \sum_{j \neq i} \mathbf{H}(cos^{-1}(\hat{f}^i, \hat{f}^j), cos^{-1}(\overline{f}^i, \overline{f}^j)) \tag{3}$$

where $\mathbf{H}$ denotes the Huber loss. During the policy distillation procedure, the weight of the centralized policy $\pi_{\text{teacher}}$ is frozen. The comparision of IDT, MADT [26] and our approach is summarized in Fig. 2

**Mapping Networks.** Inspired by recent works [4, 11, 5] which use an MLP projection head to provide flexibility for contrastive representation learning, we propose to use a pair of mapping networks $(M, N)$ to help policy distillation and prevent from loss of information induced by relational distillation loss. By leveraging mapping networks that remove information irrelevant to structural relation yet potentially useful in downstream policy learning, more information can be formed and maintained in the feature $f$. Specifically, $M$ and $N$ are simple MLPs that transform the features before relational policy distillation. Note that the weight of mapping networks are non-shared ($M \neq N$) in our setting since the teacher and student policies are inherently asymmetric in the amount of information allowed for reasoning (the teacher has access to privileged information from all agents). Therefore, the relational policy distillation becomes

$$L^i_{rel} = \sum_{j \neq i} \mathbf{H}(cos^{-1}(M(\hat{f}^i), M(\hat{f}^j)),$$

$$cos^{-1}(N(\overline{f}^i), N(\overline{f}^j)))$$

However, a learnable and non-fixed $M$ effectively forms a moving target in the objective, causing unstable learning. To stabilize the distillation process, we adopt a momentum-like update for mapping networks. We make the update frequency of the teacher's mapping network $M$ lower than the student's mapping network $N$. In other words, we update $M$ every $e$ updates of $N$. We empirically found that setting $e = 4$ improves the convergence speed, but the converged performance is insensitive to $e$.

In summary, the overall learning objective for agent $i$ is

$$L^i_{total} = L^i_{action} + \alpha L^i_{rel} + \beta L^i_{KL} \tag{4}$$

where $\alpha$ and $\beta$ are hyperparameters that determine the importance of the proposed policy distillation.

## 4 Experiments

We execute a series of experiments to evaluate whether the proposed method is effective at solving offline MARL problems. Specifically, our experiments seek to answer the following questions: first, does our method perform favorably against a wide range of existing approaches based on sequence modeling, imitation learning, and offline reinforcement learning (Section 4.1)? Specifically, we compare our method with model-free offline MARL methods based on TD-learning and MADT, a

|  | **Fill-In** | **Equal Space** | **Grid-World** | **Highway** |
|---|---|---|---|---|
| BC | $-12.43 \pm 0.21$ | $-9.35 \pm 0.64$ | $1.49 \pm 0.16$ | $13.38 \pm 1.14$ |
| IDT [3] | $-7.83 \pm 0.42$ | $-7.99 \pm 0.42$ | $1.52 \pm 0.27$ | $18.71 \pm 1.53$ |
| MADT [26] | $-6.51 \pm 0.21$ | $-6.91 \pm 0.92$ | $1.57 \pm 0.34$ | $18.78 \pm 1.27$ |
| MA-CQL [18] | $-9.41 \pm 1.72$ | $-6.99 \pm 0.38$ | $1.44 \pm 0.30$ | $17.69 \pm 1.32$ |
| MA-ICQ [45] | $-9.72 \pm 0.39$ | $-7.12 \pm 0.29$ | $1.62 \pm 0.29$ | $18.01 \pm 1.27$ |
| MA-BCQ [10] | $-8.11 \pm 0.20$ | $-7.06 \pm 0.59$ | $1.49 \pm 0.44$ | $17.92 \pm 1.48$ |
| MA-GAIL [39] | $-3.41 \pm 0.12$ | $-8.43 \pm 0.42$ | $1.51 \pm 0.32$ | $16.48 \pm 1.80$ |
| MA-AIRL [47] | $-11.41 \pm 0.07$ | $-8.43 \pm 0.63$ | $1.52 \pm 0.37$ | $16.76 \pm 1.95$ |
| Ours | $\mathbf{-3.41 \pm 0.12}$ | $\mathbf{-2.43 \pm 0.72}$ | $\mathbf{2.09 \pm 0.22}$ | $\mathbf{23.35 \pm 0.91}$ |

|  | **SMAC** [35] | | | |
|---|---|---|---|---|
|  | 2s3z | 3s5z | 8m9m | 3s5z vs 3s6z |
| BC | $14.77 \pm 1.01$ | $11.32 \pm 0.79$ | $11.45 \pm 1.14$ | $10.86 \pm 0.99$ |
| IDT [3] | $17.63 \pm 1.80$ | $15.99 \pm 1.11$ | $15.93 \pm 0.86$ | $16.33 \pm 1.93$ |
| MADT [26] | $18.09 \pm 1.26$ | $16.18 \pm 1.05$ | $17.11 \pm 1.83$ | $16.91 \pm 2.10$ |
| MA-CQL [18] | $17.04 \pm 1.38$ | $15.02 \pm 1.93$ | $14.92 \pm 1.87$ | $15.32 \pm 2.42$ |
| MA-ICQ [45] | $17.42 \pm 1.52$ | $15.36 \pm 2.01$ | $14.72 \pm 1.22$ | $14.99 \pm 2.21$ |
| MA-BCQ [10] | $17.08 \pm 1.12$ | $15.09 \pm 0.84$ | $14.32 \pm 1.02$ | $15.78 \pm 1.64$ |
| MA-GAIL [39] | $15.01 \pm 1.12$ | $13.99 \pm 0.84$ | $13.99 \pm 0.69$ | $14.98 \pm 2.04$ |
| MA-AIRL [47] | $15.11 \pm 1.12$ | $14.02 \pm 0.84$ | $14.01 \pm 0.79$ | $14.95 \pm 2.18$ |
| Ours | $\mathbf{18.12 \pm 1.31}$ | $\mathbf{16.98 \pm 1.19}$ | $\mathbf{18.33 \pm 0.99}$ | $\mathbf{18.78 \pm 2.01}$ |

Table 1: **Quantitative Results**. We show the average and standard deviation of return pre-agent, and all the experiments are merged with 10 random seeds. As for detailed information about the offline dataset, please refer to appendix D.

|  | **Grid-World** | | | **Highway** | | |
|---|---|---|---|---|---|---|
| Dataset Quality | good | normal | poor | good | normal | poor |
| BC | $1.49 \pm 0.16$ | $1.29 \pm 0.05$ | $1.01 \pm 0.12$ | $13.38 \pm 1.14$ | $10.23 \pm 0.91$ | $8.71 \pm 0.72$ |
| IDT [3] | $1.52 \pm 0.27$ | $1.45 \pm 0.12$ | $1.43 \pm 0.09$ | $18.71 \pm 1.53$ | $18.01 \pm 1.39$ | $\mathbf{17.58 \pm 1.01}$ |
| MADT [26] | $1.57 \pm 0.34$ | $1.50 \pm 0.17$ | $1.44 \pm 0.11$ | $18.78 \pm 1.27$ | $18.03 \pm 1.02$ | $17.84 \pm 0.84$ |
| MA-CQL [18] | $1.44 \pm 0.30$ | $1.40 \pm 0.21$ | $1.31 \pm 0.11$ | $17.69 \pm 1.32$ | $16.88 \pm 0.93$ | $15.98 \pm 0.67$ |
| MA-ICQ [45] | $1.62 \pm 0.29$ | $1.37 \pm 0.10$ | $1.32 \pm 0.09$ | $18.01 \pm 1.27$ | $16.54 \pm 0.74$ | $16.03 \pm 0.83$ |
| MA-BCQ [10] | $1.49 \pm 0.44$ | $1.39 \pm 0.14$ | $1.37 \pm 0.05$ | $17.92 \pm 1.48$ | $16.89 \pm 0.78$ | $15.73 \pm 0.58$ |
| MA-GAIL [39] | $1.51 \pm 0.32$ | $1.19 \pm 0.31$ | $1.16 \pm 0.28$ | $14.48 \pm 1.80$ | $11.27 \pm 1.58$ | $9.12 \pm 1.78$ |
| MA-AIRL [47] | $1.52 \pm 0.37$ | $1.21 \pm 0.27$ | $1.11 \pm 0.25$ | $16.76 \pm 1.95$ | $15.82 \pm 1.93$ | $10.22 \pm 1.87$ |
| Ours | $\mathbf{2.09 \pm 0.22}$ | $\mathbf{1.89 \pm 0.12}$ | $\mathbf{1.81 \pm 0.19}$ | $\mathbf{23.35 \pm 0.91}$ | $\mathbf{20.01 \pm 1.01}$ | $17.13 \pm 1.12$ |

Table 2: Offline learning results on offline trajectories with different quality. In general, our approach outperform other baselines when the quality of the offline dataset is not perfect.

concurrent work that also solves MARL via sequence modeling. Next, we conduct careful ablation studies to evaluate the contribution of each component within our framework (Section 4.2). To test the sample efficiency of various approaches, we further show the performance of various methods when different numbers of demonstrations are provided (Section 4.4). Finally, we analyze the convergence rate (Section 4.5) and discuss the scalability (Section 5) of our method.

**Baselines.** We compare our approach with three group of baselines including sequence modeling (**IDT**, **MADT**), offline RL (**MA-CQL**, **MA-ICQ**, **MA-BCQ**) and imitation learning/inverse RL (**BC**, **MA-GAIL**, **MA-AIRL**), and we briefly introduce them as follow.

**BC**: behavior cloning. **IDT**: each agent is represented by a decision transformer [3] independently. **MADT** [26]: a weight-sharing decision transformer is used to represent each agent's policy. **MA-CQL** [18]: conservative Q-learning (CQL), which aims to address the extrapolation error by learning a conservative Q-function such that the expected value of a policy under this Q-function lower-bounds its true value. **MA-ICQ** [45]: implicit constraint Q-learning (ICQ), which improves upon MA-CQL by only trusting the state-action pairs given in the dataset for value estimation under the multi-agent setting. **MA-BCQ** [10]: batch-constrained reinforcement learning, which restricts the action space in order to force the agent towards behaving close to on-policy with respect to a subset of the given data. **MA-GAIL** [39]: an inverse reinforcement learning for multi-agent scenario. **MA-AIRL** [47]: a framework for multi-agent inverse reinforcement learning, which is effective and scalable for Markov games with high-dimensional state-action space and unknown dynamics. For detail experimental setting about our approach and baselines, please refer to appendix A.

**Environments.** We test our approach on three multi-agent environments. **Fill-In**: a grid-world environment that agents are required to pass all the blocks in map. **Equal Space**: a particle system environment that agents need to keep the same distance between each other. **SMAC** [28]: a collaborative multi-agent reinforcement learning based on Blizzard's StarCraft II RTS game, and we choose four different scenarios, 2s3z, 3s5z, 8m9m, and 3s5z vs 3s6z. **Grid-World**: a simple grid-world environment that allow agents to conduct discrete actions to move and collect the corresponding objects. **Highway**: a multi-agent environment that simulate highway traffic scenario. Each controllable agent drives a car and the goal is to reach a high speed while not colliding into neighboring vehicles. For detail experimental setting, please refer to appendix B.

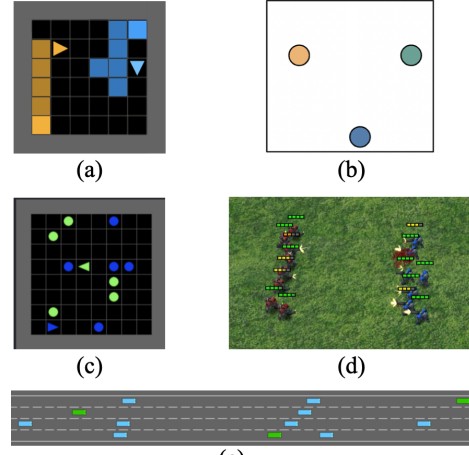

(a)   (b)

(c)   (d)

(e)

Figure 4: **Environments** we used in our experiments: (a) Fill-In (b) Equal Space (c) Grid-World (d) SAMC (e) Highway. See the detailed description of each task in Section 4.

**Offline Dataset** The offline MARL dataset for $n$ agents is represented as a set of trajectories $\tau := (o_t^{1:n}, a_t^{1:n}, r_t^{1:n})_{t=1}^T$. For Grid-World and Highway, we train a centralized policy with PPO [36] and treat it as the expert to generate demonstrated trajectories. The offline datasets for SMAC [35] are collected by running a policy trained with MAPPO [46]. We also provide more information about datasets in appendix C.

| | Fill-In | Equal Space | Grid-World | Highway |
|---|---|---|---|---|
| Ours (conventional distillation) | $-4.11 \pm 0.22$ | $-5.43 \pm 0.51$ | $1.62 \pm 0.29$ | $18.33 \pm 1.27$ |
| Ours ($M = N$) | $-3.11 \pm 0.22$ | $-2.61 \pm 0.33$ | $1.42 \pm 0.19$ | $14.92 \pm 1.87$ |
| Ours - Momentum Update | $-3.92 \pm 0.31$ | $-2.44 \pm 0.53$ | $1.79 \pm 0.19$ | $18.62 \pm 1.98$ |
| Ours - Mapping Network | $\mathbf{-3.32 \pm 0.42}$ | $-2.87 \pm 0.19$ | $1.88 \pm 0.11$ | $18.21 \pm 1.99$ |
| Ours - KL Distillation | $-3.89 \pm 0.33$ | $-2.99 \pm 0.27$ | $1.99 \pm 0.51$ | $20.11 \pm 1.88$ |
| Ours - KL Distillation
    - Mapping Network | $-3.30 \pm 0.21$ | $-2.71 \pm 0.42$ | $1.59 \pm 0.44$ | $18.40 \pm 1.09$ |
| Ours Full method | $-3.41 \pm 0.12$ | $\mathbf{-2.43 \pm 0.72}$ | $\mathbf{2.09 \pm 0.22}$ | $\mathbf{23.35 \pm 0.91}$ |

| | SMAC [35] | | | |
|---|---|---|---|---|
| | 2s3z | 3s5z | 8m9m | 3s5z vs 3s6z |
| Ours (conventional distillation) | $16.11 \pm 1.87$ | $15.78 \pm 1.78$ | $15.99 \pm 1.11$ | $17.93 \pm 1.32$ |
| Ours ($M = N$) | $14.14 \pm 1.31$ | $12.78 \pm 1.08$ | $15.40 \pm 2.11$ | $12.27 \pm 1.11$ |
| Ours - Momentum Update | $15.69 \pm 1.91$ | $15.20 \pm 0.81$ | $16.39 \pm 2.70$ | $17.51 \pm 1.49$ |
| Ours - Mapping Network | $15.23 \pm 1.81$ | $15.01 \pm 0.31$ | $16.87 \pm 0.91$ | $17.01 \pm 0.32$ |
| Ours - KL Distillation | $16.32 \pm 0.91$ | $16.21 \pm 0.55$ | $17.01 \pm 1.26$ | $17.80 \pm 1.08$ |
| Ours - KL Distillation
    - Mapping Network | $15.01 \pm 1.23$ | $14.98 \pm 1.06$ | $16.42 \pm 1.22$ | $17.71 \pm 1.88$ |
| Ours Full method | $\mathbf{18.12 \pm 1.31}$ | $\mathbf{16.98 \pm 1.19}$ | $\mathbf{18.33 \pm 0.99}$ | $\mathbf{18.78 \pm 2.01}$ |

Table 3: Ablation study. We show the average and standard deviation of return pre-agent, and all the experiments are merged with 10 random seeds.

## 4.1 Quantitative Results

We present the quantitative results in Table 1. For all the experiments, we report the mean and standard deviation of rewards based on 10 runs using different random seeds. The results show that our approach outperforms other baselines in all three environments. In general, we find that offline RL approaches reach higher performance compared with imitation learning and inverse RL methods. It is because offline RL approach can use reward signals to judge the quality of the policy and try to infer optimal strategy. Sequence modeling methods slightly outperform offline RL ones since Sequence modeling methods allow us to bypass bootstrapping and perform credit assignments directly via self-attention. Besides, comparing ours and IDT, we can find that the proposed structural knowledge distillation does provide performance benefits.

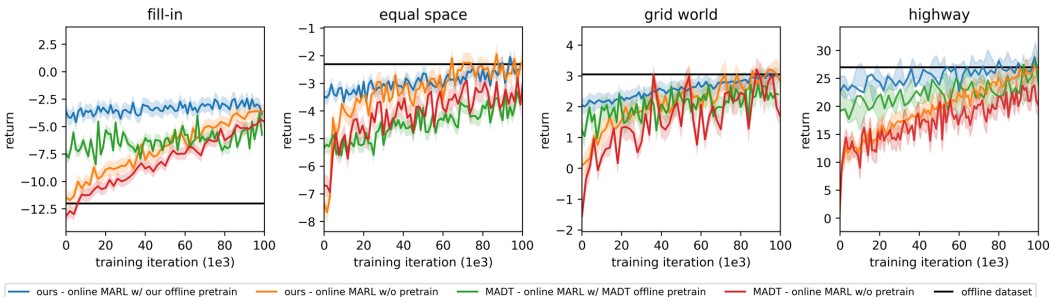

Figure 6: Learning curves of online finetuning. The experimental results are merged with 10 random seeds, and the shaded area represent the standard deviation.

We also find that our approach performs better than inverse MARL methods. Although inverse MARL discards the necessity of reward signal, it assumes the quality of demonstration trajectories is perfect. Under this assumption, it is possible to infer reward signals in an adversarial manner. However, since it's hard to guarantee the quality of multi-agent trajectories due to its dependency on other agents' behaviors, the drawback of inverse MARL becomes obvious.

In Table 2, we present the quantitative results with different qualities of offline demonstrations and find that our method outperforms all baselines except for one setting. The results suggest that our method is more robust to various demonstration qualities across tasks. We hypothesize our method performs on par with baselines on *Highway-poor* because learning a strong centralized decision transformer from poor demonstrations is challenging. This in turns affects the performances of student policies.

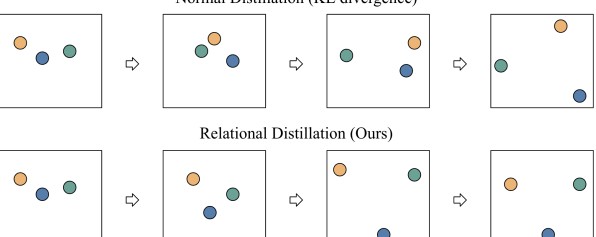

Figure 5: **Qualitative results** on equal space environment. By observing the behavior of the agents, our approach encourage agents to behave more efficiently.

## 4.2 Ablation Study

To validate the effectiveness of each component in our approach. To be more specific, we compare our approach with following variant. **Ours - Momentum Update**: the variant that both mapping network ($M$ and $N$) are updated with the same frequency. **Ours - Mapping Network**: the variant that relation knowledge distillation is directly applied to feature space of decision transformer. **Ours - KL divergence**: our approach without the KL divergence between teacher and student as distillation. **Ours (conventional distillation)**: our approach but use KL divergence for knowledge distillation from the teacher policy. **Ours** ($M = N$): the variant of our approach that the mapping networks for teacher policy and student policies share the same weights.

The results show that all the components in our approach are required. Specifically, **Ours - Mapping Network** achieves poor performance on 7 out of 8 tasks. It shows that Mapping Networks that transform the students' and teacher's features into the same space for distillation are essential. Additionally, according to the standard deviation of the performance, we find that approaches that do not use momentum update are less reliable. KL divergence also provide performance benefit, but not significant compared with mapping networks and momentum update.

We also find that **ours (conventional distillation)** performs the worst, which highlight that the proposed our knowledge distillation is more effective than transitional knowledge distillation. It is because structural relation among multi-agent can potentially represent how agent interact with each other, which is the critical signal for a multi-agent task. Fig. 5 shows the behavior of the policies trained by our approach and ours with conventional distillation, respectively. To make agents keep the same distance from each other, policy obtained by our method tends to make agent move the the closest corner and form an equilateral triangle structure. On the other hand, policy obtained by offline learning and conventional distillation encourage specific agents to go to specific corners, which is a sub-optimal solution. The performance **Ours** ($M = N$) drops a lot, which represents that making the weights of the two mapping networks non-shared is a reasonable choice.

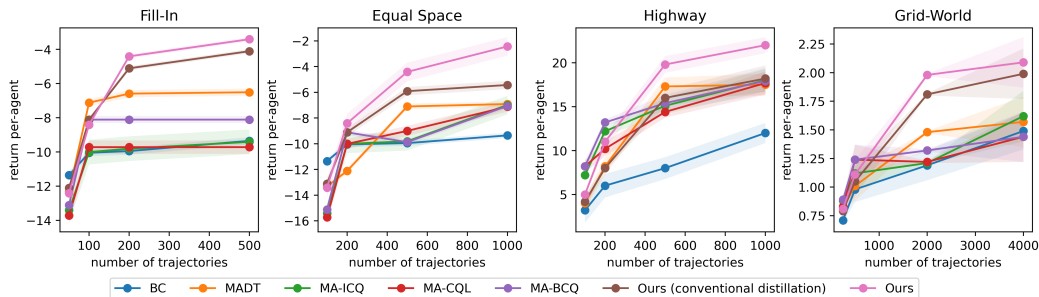

Figure 7: Data efficiency in offline learning setting on highway and grid-world environment. The experimental results are merged with 10 random seeds, and the shaded area represents the standard deviation. In general, our approach can have better data efficiency.

### 4.3 Finetuning

To verify whether the offline pre-trained framework can be further improved, we finetune the pre-trained framework with MAPPO [46]. In Fig 6, we show the comparison of learning curves between finetuning based on the pre-trained policy and policy trained from scratch. The experimental results show that the pre-trained policy can get improvement with an online multi-agent reinforcement learning approach.

### 4.4 Data Efficiency

We also show the performance with a different number of trajectories in Fig. 7 . The results show that our relational knowledge distillation achieves better performance than conventional knowledge distillation when the number of offline trajectories is relatively low. Besides, we also find that offline RL approaches perform well with limited offline trajectories compared with our approach. We hypothesize that the cause is that the transformer-based approach requires a large amount of data. We stress that a common offline MARL setting allows abundant yet non-interactive demonstration and thus our approach is still meaningful and desirable. Finally, we find that naive imitation learning and behavior cloning perform the worst in all the cases, and we hypothesize that imitation learning suffers from covariate shift, which can be more severe when the training data is few and unable to cover a sufficiently large domain.

### 4.5 Convergence Rate

We observe better or comparable convergence with the centralized decision transformer for all environments compared to MADT, e.g., in SMAC 3s5z vs 3s6z with poor quality data, the centralized transformer is ∼8% better at 20k iterations (not fully converged). For knowledge distillation, it requires roughly similar total training iterations for the student policies to converge. Besides, adding a centralized decision transformer doesn't introduce a large overhead in training time. In general, adding one more centralized DT only requires less than an additional 10% training time in our experiment, the training time as well as learning curves for each experiment are presented in Appendix B.

## 5 Limitation and Broader Impact

The scalability of multi-agent frameworks is important. Our approach depends on a centralized decision transformer, which predicts all actions of each agent based on the returns and actions of all the agents. Therefore, the centralized decision transformer may fall short in scalability. However, our experiments find that we can still reach comparable performance compared with MADT when the number of agents increases. We leave scalability in multi-agent policy distillation for future work. Additionally, we note that offline MARL algorithms may lead to negative applications such as flying a fleet of drones for surveillance.

## 6 Conclusion

In this work, we propose an offline MARL algorithms that first trains a teacher policy as if the dataset is generated by a single agent. Once the teacher policy is trained, we use its predictions to supervise the student using a novel distillation objective that considers agents' relations.

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
