# A Background

## A.1 Partially Observable Mackov Decision Process

We follow previous works [25] to consider MARL as a partially observable Markov games [22]. We define a set of states $\mathbf{S}$ describing the possible configurations of all $n$ agents. For each agent $i$, we define a set of actions $\mathbf{A}^i$ and a set of local observations $\mathbf{O}^i$. A multi-agent task can then be described as $\mathbf{T} = \{\mathbf{S}, \mathbf{A}^{1:n}, \mathbf{O}^{1:n}, n\}$.

At each step of the Markov game, each agent $i$ uses a stochastic policy $\pi^i : \mathbf{O}^i \times \mathbf{A}^i \to [0, 1]$ to select an action which leads to the next state according to the dynamics function $D : \mathbf{S} \times \mathbf{A}^1 \times ... \times \mathbf{A}^n \to \mathbf{S}$. Then, each agent $i$ gets rewards as a function of the state and agent's action $r^i : \mathbf{S} \times \mathbf{A}^i \to \mathbf{R}$, and receives an observation correlated with the state $o^i : \mathbf{S} \to \mathbf{O}^i$. The initial states are defined by a distribution $\sigma : \mathbf{S} \to [0, 1]$. Each agent $i$ intends to maximize their own total expected return $R^i = \sum_{t=1}^{T} \gamma_t r_t^i$ where $\gamma$ is a discount factor and $T$ stands for the maximum of steps an agent can take. In the following paragraph, we use superscript to indicate agent's index and subscript to indicate time step for states, observations, rewards and actions.

## A.2 Decision Transformer

Decision Transformer [3] using Transformer [44] which is an architecture to efficiently model sequential data shows its ability to cast the problem of RL as conditional sequence modeling.

The core component of transformer is attention mechanism [44]. Let $\mathbf{Q} \in R^{m_q \times d_k}$ be the queries, $\mathbf{K} \in R^{m_k \times d_k}$ and $\mathbf{V} \in R^{m_k \times d_v}$ where $m_*$ represents element numbers of different inputs and $d_*$ represents the corresponding element dimensions. The output of the attention mechanism are computed as

$$Atten(\mathbf{Q}, \mathbf{K}, \mathbf{V}) = softmax(\frac{\mathbf{Q}\mathbf{K}^T}{\sqrt{d_k}})\mathbf{V} \tag{5}$$

The attention mechanism can be extend to multi-head attention, which provides capability to jointly attend to information from different representation subspaces. Its formulation is shown as follow

$$MultiHead(\mathbf{Q}, \mathbf{K}, \mathbf{V}) = Concat(head_1, ..., head_h)\mathbf{W}^O \tag{6}$$

$$head_i = Atten(\mathbf{Q}\mathbf{W}_i^Q, \mathbf{K}\mathbf{W}_i^K, \mathbf{V}\mathbf{W}_i^V) \tag{7}$$

Besides, the position-wise feed-forward network is another critical module in the transformer. It consists of two linear transformations with a ReLU activation in between. $d_{model}$ is the dimensionality of inputs and outputs, and $d_{ff}$ is that of the feed forward layer, so the position-wise feed-forward network becomes

$$FFN(x) = max(0, x\mathbf{W}_1 + \mathbf{b}_1)\mathbf{W}_2 + \mathbf{b}_2 \tag{8}$$

Across different positions are the same linear transformations with shared weights. Note that the position encoding for leveraging the order of the sequence as follows:

$$PE(pos, 2i) = \sin pos/10000^{2i/d_{model}}$$
$$PE(pos, 2i+1) = \cos pos/10000^{2i/d_{model}} \tag{9}$$

By conditioning an autoregressive model on the desired return (reward), past states, and actions that implicitly constructs state-return associations via similarity of key and query vectors, the Decision Transformer model can generate future actions that achieve the desired return.

# B Implementation Details

Our method are describe in the main paper Sec. 3, and we present the hyperparameter and detail architecture. The source code will be public after the acceptance. The proposed framework is trained on the servers with NVIDIA RTX 2080Ti GPUs and AMD 2990WX CPU. We also provide learning curves for our offline learning setting in Fig. 8

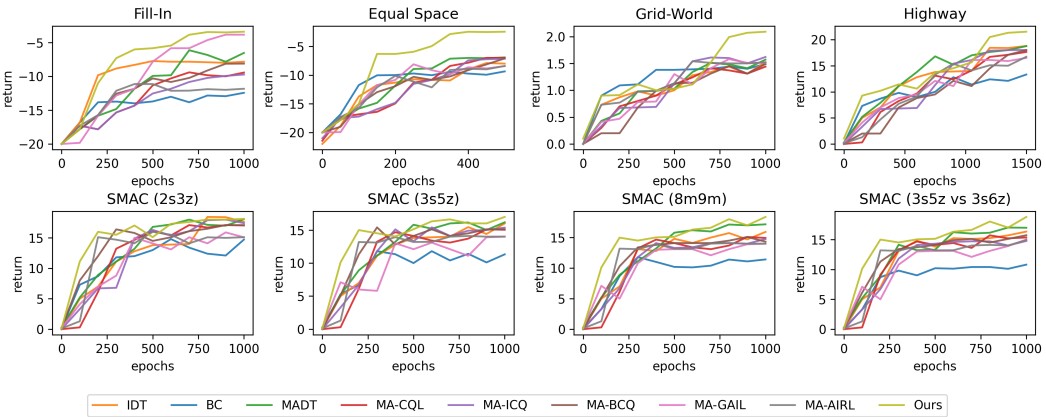

Figure 8: **Learning curves** in our offline learning setting

## B.1 Independent Decision Transformer (IDT)

The implementation of decision transformer [3] is adapted from the official released codebase[2]. For each agent, we construct a separated decision transformer learned from trajectory modeling as the policy to control the agent, and the backbone of the transformer is similar to GPT-2 [31]. We summarize the hyperparameters used in our experiments in the table below.

| | **Fill-In** | **Equal Space** | **Grid-World** | **Highway** | **SMAC** [35] | | | |
|---|---|---|---|---|---|---|---|---|
| | | | | | 2s3z | 3s5z | 8m9m | 3s5z vs 3s6z |
| offline lr | 0.0001 | 0.0001 | 0.0001 | 0.0001 | 0.0001 | 0.0001 | 0.0001 | 0.0001 |
| weight decay | 0.0001 | 0.0001 | 0.0001 | 0.0001 | 0.0001 | 0.0001 | 0.0001 | 0.0001 |
| offline epochs | 1000 | 500 | 1000 | 1500 | 1000 | 1000 | 1000 | 1000 |
| training time (hr) | 5.4 | 3.3 | 6.1 | 8.4 | 10.1 | 13.5 | 14.12 | 16.3 |

Table 4: Hyperparameters for IDT [3]

## B.2 Behavior Cloning (BC)

The implementation of behavior cloning is pretty simple. We create a 3-layer MLP with hidden dimensions 128 as the policy we would like to learn. The policy network takes a series of local observations as input and predict action distribution, and it is trained in regressing manner. We summarize the hyperparameters used in our experiments in the table below.

| | **Fill-In** | **Equal Space** | **Grid-World** | **Highway** | **SMAC** [35] | | | |
|---|---|---|---|---|---|---|---|---|
| | | | | | 2s3z | 3s5z | 8m9m | 3s5z vs 3s6z |
| offline lr | 0.0005 | 0.0005 | 0.0002 | 0.0002 | 0.0002 | 0.0002 | 0.0002 | 0.0002 |
| weight decay | 0.0001 | 0.0001 | 0.0001 | 0.0001 | 0.0001 | 0.0001 | 0.0001 | 0.0001 |
| offline epochs | 1000 | 500 | 1000 | 1500 | 1000 | 1000 | 1000 | 1000 |
| training time (hr) | 2.8 | 1.9 | 3.3 | 4.1 | 7.3 | 12.8 | 13.1 | 13.1 |

Table 5: Hyperparameters for Behavior Cloning.

## B.3 Multi-Agent Decision Transformer (MADT)

The implementation of MADT [26] is leveraged from their offical implementation [3]. The backbone of the transformer is also similar to GPT-2 [31]. To be more specific, it contains two-layer transformer blocks. We summarize the hyperparameters used in our experiments in the table below.

---

[2]`https://github.com/kzl/decision-transformer` with MIT license

[3]`https://openreview.net/attachment?id=WO8IqLMlMer&name=supplementary_material`

| | Fill-In | Equal Space | Grid-World | Highway | SMAC [35] | | | |
|---|---|---|---|---|---|---|---|---|
| | | | | | 2s3z | 3s5z | 8m9m | 3s5z vs 3s6z |
| offline lr | 0.0001 | 0.0001 | 0.0001 | 0.0001 | 0.0001 | 0.0001 | 0.0001 | 0.0001 |
| weight decay | 0.0001 | 0.0001 | 0.0001 | 0.0001 | 0.0001 | 0.0001 | 0.0001 | 0.0001 |
| offline epochs | 1000 | 500 | 1000 | 1500 | 1000 | 1000 | 1000 | 1000 |
| training time (hr) | 5.2 | 3.2 | 6.2 | 8.6 | 10.1 | 13.0 | 14.2 | 15.9 |

Table 6: Hyperparameters for MADT [26]

## B.4 Multi-Agent Conservative Q-Learning (MA-CQL)

The implementation is adapted from the official implementation of CQL [18][4] and MADT [26]. The Q-network is formed 3-layer MLP with hidden dimensions 256. A mixing network [32] used to fuse the q value estimation for the agents is applied. The architecture of mixing network is 2-layer MLP with hidden dimension 64.

| | Fill-In | Equal Space | Grid-World | Highway | SMAC [35] | | | |
|---|---|---|---|---|---|---|---|---|
| | | | | | 2s3z | 3s5z | 8m9m | 3s5z vs 3s6z |
| offline lr | 0.0001 | 0.0001 | 0.0001 | 0.0001 | 0.0001 | 0.0001 | 0.0001 | 0.0001 |
| offline batch size | 128 | 128 | 128 | 128 | 128 | 128 | 128 | 128 |
| weight decay | 0.0001 | 0.0001 | 0.0001 | 0.0001 | 0.0001 | 0.0001 | 0.0001 | 0.0001 |
| offline epochs | 1000 | 500 | 1000 | 1500 | 1000 | 1000 | 1000 | 1000 |
| training time (hr) | 5.9 | 3.5 | 6.4 | 9.1 | 10.4 | 13.1 | 15.1 | 15.9 |

Table 7: Hyperparameters for MA-CQL [18]

## B.5 Multi-Agent Implicit Constraint Q-Learning (MA-ICQ)

The implementation is adapted from the official implementation of ICQ [45][5] and MADT [26]. The two Q-network are formed 3-layer MLP with hidden dimensions 256, respectively. A mixing network [32] used to fuse the q value estimation for the agents is applied. The architecture of mixing network is 2-layer MLP with hidden dimension 64.

| | Fill-In | Equal Space | Grid-World | Highway | SMAC [35] | | | |
|---|---|---|---|---|---|---|---|---|
| | | | | | 2s3z | 3s5z | 8m9m | 3s5z vs 3s6z |
| offline lr | 0.0001 | 0.0001 | 0.0001 | 0.0001 | 0.0001 | 0.0001 | 0.0001 | 0.0001 |
| offline batch size | 128 | 128 | 128 | 128 | 128 | 128 | 128 | 128 |
| weight decay | 0.0001 | 0.0001 | 0.0001 | 0.0001 | 0.0001 | 0.0001 | 0.0001 | 0.0001 |
| offline epochs | 1000 | 500 | 1000 | 1500 | 1000 | 1000 | 1000 | 1000 |
| training time (hr) | 4.8 | 3.0 | 5.9 | 7.4 | 9.3 | 11.9 | 14.2 | 14.4 |

Table 8: Hyperparameters for MA-ICQ [45]

## B.6 Multi-Agent Batch-Constrained Deep Q-Learning (MA-BCQ)

The implementation is adapted from the official implementation of BCQ [10][6] and MADT [26]. The two Q-networks are formed 3-layer MLP with hidden dimensions 256. A mixing network [32] used to fuse the q value estimation for the agents is applied. The architecture of mixing network is 2-layer MLP with hidden dimension 64.

## B.7 Ours

The implementation of our approach is adapted from from the official released codebase[7] of decision transformer [3]. As for online finetuning, the finetuning procedure is as the same as that of MADT [26]. The optimization algorithm used in finetuning phase is MAPPO [46]. We summarize the hyperparameters for both offline and online learning in the table below.

---

[4] https://github.com/aviralkumar2907/CQL.git

[5] https://github.com/YiqinYang/ICQ.git

[6] https://github.com/sfujim/BCQ with MIT license

[7] https://github.com/kzl/decision-transformer

| | Fill-In | Equal Space | Grid-World | Highway | SMAC [35] | | | |
|---|---|---|---|---|---|---|---|---|
| | | | | | 2s3z | 3s5z | 8m9m | 3s5z vs 3s6z |
| offline lr | 0.0001 | 0.0001 | 0.0001 | 0.0001 | 0.0001 | 0.0001 | 0.0001 | 0.0001 |
| offline batch size | 128 | 128 | 128 | 128 | 128 | 128 | 128 | 128 |
| weight decay | 0.0001 | 0.0001 | 0.0001 | 0.0001 | 0.0001 | 0.0001 | 0.0001 | 0.0001 |
| offline epochs | 1000 | 500 | 1000 | 1500 | 1000 | 1000 | 1000 | 1000 |
| training time (hr) | 5.1 | 3.1 | 6.1 | 7.8 | 9.3 | 10.9 | 12.4 | 15.2 |

Table 9: Hyperparameters for MA-BCQ [10]

| | Fill-In | Equal Space | Grid-World | Highway | SMAC [35] | | | |
|---|---|---|---|---|---|---|---|---|
| | | | | | 2s3z | 3s5z | 8m9m | 3s5z vs 3s6z |
| offline lr | 0.0001 | 0.0001 | 0.0001 | 0.0001 | 0.0001 | 0.0001 | 0.0001 | 0.0001 |
| online lr | 0.0001 | 0.0001 | 0.0001 | 0.0001 | - | - | - | - |
| weight decay | 0.0001 | 0.0001 | 0.0001 | 0.0001 | 0.0001 | 0.0001 | 0.0001 | 0.0001 |
| offline epochs | 1000 | 500 | 1000 | 1500 | 1000 | 1000 | 1000 | 1000 |
| $\alpha$ | 0.01 | 0.01 | 0.01 | 0.01 | 0.01 | 0.01 | 0.01 | 0.01 |
| $\beta$ | 0.001 | 0.002 | 0.002 | 0.001 | 0.0001 | 0.0001 | 0.0001 | 0.0001 |
| $e$ | 5 | 5 | 5 | 5 | 4 | 4 | 4 | 4 |
| online epochs | 10 | 10 | 10 | 50 | - | - | - | - |
| training time (hr) | 5.5 | 3.3 | 6.7 | 9.0 | 10.9 | 14.2 | 15.2 | 17.1 |

Table 10: Hyperparameters for our approach.

## C Environmental Setting

**Fill-In**. This environment is adapted from `https://github.com/ArnaudFickinger/gym-multigrid`. The goal of this task to to make agents fill all the blocks in the map. Therefore, a good strategy for agents to do so is that agents try to move around and avoid pass through other agents' path. The detailed environment definition is shown as follow.

The observation $o$ of the agent in the environment the 3x5 visible region, which can be converted to a small image. In the map, different agents are represented by different colors, and the passed blocks are represented by the color as the same as the passing agent with transparency. The action $a$ of the agent in the environment is the discrete action which contains [move forward, turn left, turn right, no move] to reach the adjacent partition. The reward $r$ of the agent consider how many blocks the agent already passes and how many blocks that aren't passed yet. Therefore, the reward can formulated as follow $r = -\#$ of not filled space $+ \#$ of slots occupied by specific agent. Note that once a block is passed by an agent, another agent passing the block won't be rewarded.

**Equal Space**. The environment is adapted from `https://github.com/openai/multiagent-particle-envs`. The goal of this task is that agents need to keep the same distance between each other, and there are three agents in the environment. Ideally, the agents should form a triangle in the end, so all the distances between the agents are the same. Here is the detailed environment definition.

The observation $o$ is the relative position of other agents. The action $a$ of the agent in the environment is the continuous action which contains $[\delta_x, \delta_y]$ allowing agent move in the environment.

**Grid-World**. This environment is adapted from `https://github.com/ArnaudFickinger/gym-multigrid`.

The observation $o$ of the agent in the environment the 5x5 visible region. To be more specific, 4 10x10 maps identify the other agent, apples, obstacles and target position, respectively. The action $a$ of the agent in the environment is the discrete action which contains [move forward, turn left, turn right, no move] to reach the adjacent partition. The reward $r$ represents whether the agent collects the object. To be more specific, $r$ is 0.5 if the agent gets the object, otherwise, $r$ is 0.

**Highway**. The environment is leveraged from `https://github.com/eleurent/highway-env`. The objective of this task is to make cars move as fast as possible and prevent collision at the same time. To be more specific, there are three cars to be control and other cars are controlled by build-in controller. Therefore, a good strategy for agents to do so is that agents try to accelerate and switch lane to pass through other cars if other cars move too slowly. Here is the detailed environment definition.

The observation $o$ of the agent in the environment consider the status of the controlled car and other close cars. The status of a car can be represented as follow $F = [L, x_{pos}, y_{pos}, v_x, v_y]$, where $L$ means whether a car is still alive, $x_{pos}$ and $y_{pos}$ indicate the position of the car, $v_x$ and $v_y$ represent the velocity of the car. Therefore, the observation of the agent is $o = [F_{self}, F_{close}]$ The action $a$ of the agent in the environment is [no operation, slow down, accelerate, switch to left lane, switch to right lane]. The reward $r$ consider whether collision happens and how fast the car moves. Specifically, the reward is defined as $r = 25 \cdot r_{collision} + 0.2 \cdot r_{high\_speed}$. $r_{high\_speed}$ represents the speed of the car. $r_{collision}$ is -1 if the car collide with other car, otherwise, it is 0.

**SMAC** [35]. SMAC is leveraged from `https://github.com/oxwhirl/smac` which is WhiRL's environment for research in the field of collaborative multi-agent reinforcement learning (MARL) based on Blizzard's StarCraft II RTS game. SMAC makes use of Blizzard's StarCraft II Machine Learning API and DeepMind's PySC2 to provide a convenient interface for autonomous agents to interact with StarCraft II, getting observations and performing actions. Unlike the PySC2, SMAC concentrates on decentralised micromanagement scenarios, where each unit of the game is controlled by an individual RL agent.

| Dataset Quality | Grid-World | | | Highway | | |
|---|---|---|---|---|---|---|
| | good | normal | poor | good | normal | poor |
| number of trajectories | 4000 | 4000 | 4000 | 1000 | 1000 | 1000 |
| return distribution | $3.02 \pm 0.29$ | $2.50 \pm 0.12$ | $2.13 \pm 0.09$ | $26.89 \pm 1.92$ | $20.91 \pm 1.39$ | $18.71 \pm 1.53$ |

Table 11: Information of the offline dataset of **Grid-World** and **Highway** used in our experiments

| | Fill-In | Equal Space | SMAC | | |
|---|---|---|---|---|---|
| | | | 2s3z | 3s5z | 3s5z vs 3s6z |
| number of trajectories | 1000 | 1000 | 4177846 | 1448424 | 1542571 |
| return distribution | $1.52 \pm 0.27$ | $-2.25 \pm 0.12$ | $19.93 \pm 0.09$ | $18.45 \pm 2.03$ | $18.35 \pm 2.04$ |

Table 12: Information of the offline dataset of **Fill-In**, **Equal Space** and **SMAC** used in our experiments

## D   Properties of Offline Dataset

We summarize the quality and amount of offline dataset in Table C and Table C. As for how the offline datasets are constructed, we illustrate them below.

**Fill-In**. The offline dataset is generated by agents randomly walking in the grid-world environment. Therefore, the quality of the offline trajectories is limited.

**Equal Space**, **Grid-World**, **Highway**. The offline dataset is generated by an expert policy trained with PPO [36]. The expert policy take observations from all the agents as input and predict actions of all the agents. In other words, it is a centralized policy.

In Gird-World and Highway, the "good" trajectories are collected with expert policy that is trained to achieve saturated performance. The expert policy takes observations from all the agents in the environment as input and predicts the actions of all the agents. The "normal" trajectories are collected with policy that reaches around 80%performance of the expert policy, and the "poor" trajectories are collected with policy achieving around 65% performance of the expert policy.

**SMAC** The dataset is provided from MADT [26]. According to MADT, the offline trajectories are collected by running a policy trained with MAPPO [46].

## E   Qualitative Results

First, we demonstrate the map after the agents conduct several action steps in **Fill-In** environment in Fig. 9. Obviously, we can find that our approach can encourage agents fill more blocks than other methods. In contrast, BC make agents randomly walk in the map since the quality of offline dataset is limited. As for IDT, we can find that agents somehow understand how to fill the blocks. However, the trajectories of the agents have some overlap, which is not a efficient strategy. MADT performs better than IDT and BC by filling more blocks, but still can't outperform our approach.

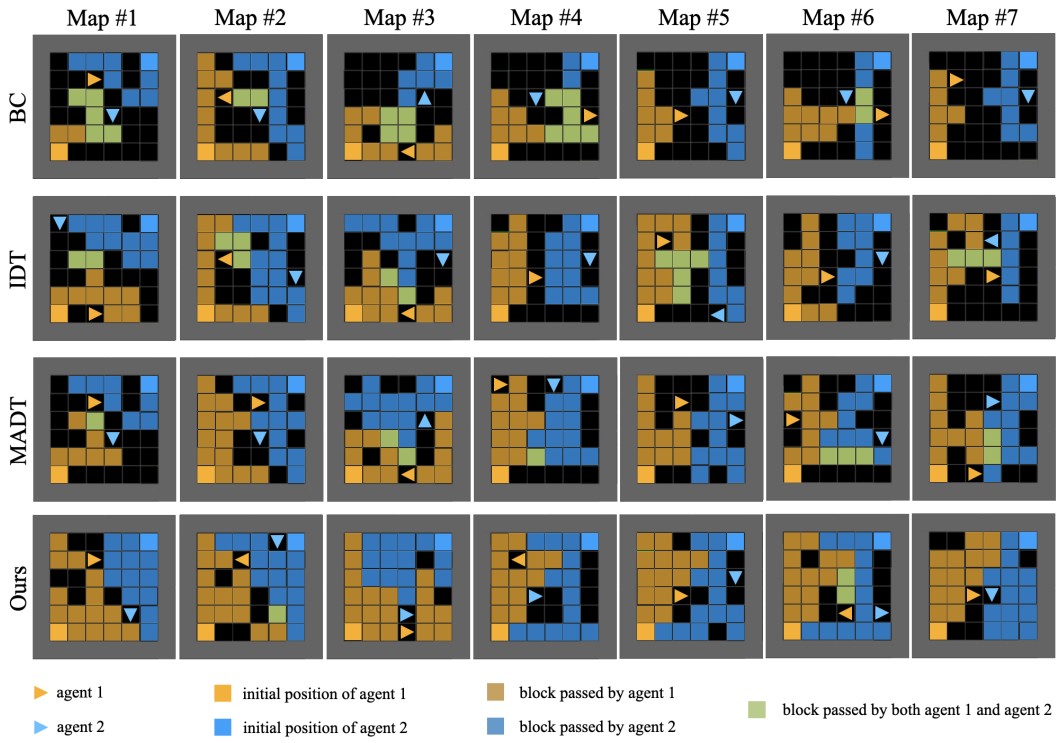

Figure 9: **Qualitative results** on **Fill-In** environment.

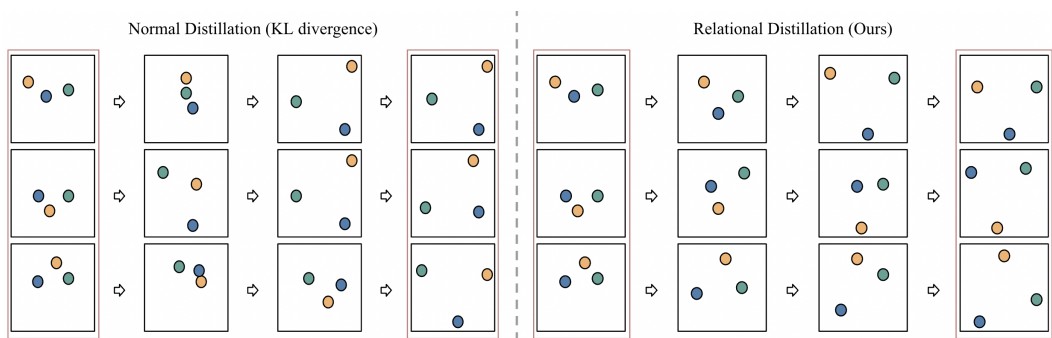

Figure 10: **Qualitative results** on equal space environment. By observing the behavior of the agents, our approach encourage agents to behave more efficiently.

Then, we show more qualitative of **Equal Space** results in 10. It shows the behavior of the policies trained by our approach and ours with conventional distillation, respectively. To make agents keep the same distance from each other, policy obtained by our method tends to make agent move the the closest corner and form an equilateral triangle structure. On the other hand, policy obtained by offline learning and conventional distillation encourage specific agents to go to specific corners, which is a sub-optimal solution.

## F Additional Experiments

### F.1 Why we use Huber loss

In Eq. 3, we use Huber loss the compare the angles between features in student policy and teacher policy. A naive alternative is use mean-square error (MSE) to measure the difference of the angles. Our motivation to use Huber loss is that the MSE has the disadvantage that it has the tendency to

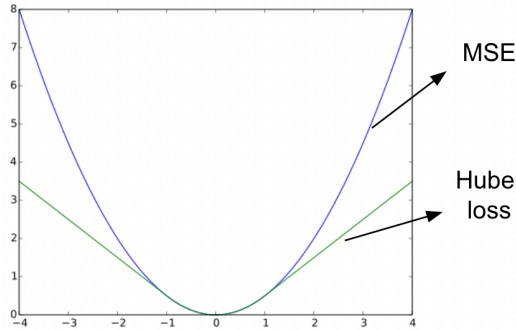

Figure 11: MSE and Hube loss. Huber loss is more stable to outliers since it grow linearly for large values.

be dominated by outliers. In contrast, Huber loss is quadratic for small values and linear for large values, so it is more stable to outliers. (see F.1) In other words, using Huber loss instead of MSE can improve the learning stablity of our approach. To further show the benefit of using Huber loss, we show the variant of using MSE to estimate the difference of angles in Table 13.

|  | Fill-In | Equal Space | Grid-World | Highway |
|---|---|---|---|---|
| Ours (Huber loss) | $-3.41 \pm 0.12$ | $-2.43 \pm 0.72$ | $2.09 \pm 0.22$ | $23.35 \pm 0.91$ |
| Ours (MSE) | $-8.21 \pm 2.32$ | $-4.43 \pm 0.92$ | $1.09 \pm 0.49$ | $19.35 \pm 2.75$ |

Table 13: Comparison of using Huber loss and MSE in our approach. The performance is merged with experiments at 10 different random seeds.