# OpenReview forum: "Offline Multi-Agent Reinforcement Learning with Knowledge Distillation"
_NeurIPS.cc/2022/Conference — NeurIPS 2022 Accept_

### Official Review · Reviewer_Vg5F · 2022-07-11

**Rating:** 6
**Confidence:** 4
**Soundness:** 3 good
**Presentation:** 3 good
**Contribution:** 4 excellent

**Summary:**

The authors aim to tackle the multi-agent offline RL domain by leveraging the power of transformers for credit assignment using sequence modeling, hence removing the reliance on TD-learning-based bootstrapping methods. The proposed method first trains a “teacher” decision transformer on the full dataset to learn cross-agent interactions. Then the teacher distills relevant knowledge into the “student agents” via “structural relation loss.” Thus the proposed method utilizes the attention mechanism for learning interactions among the agents better than the current offline RL methods.

**Questions:**

1. In the Algorithm, shouldn't the student-agent gradient update equation (φi = φi − α ∇φi (Li action + αLi rel + βLi KL)) come inside the inner for-loop?
2. I am not sure how the **Fill-In environment** works. If agent 1 has passed a particular block, will that block be then inaccessible for agent 2, and vice-versa?
3. **If the answer to the above is yes, then skip this point.** But if it is not, it is unclear why the agents here even require cross-agent credit assignment during training?
4. Re: Table 11 - Why does the return distribution for the Highway dataset have the opposite order, i.e., poor > normal > good? It seems like a typo to me.
5. Referring to Line 563 - why are obstacles and apples needed in the Fill-In environment? Is this a typo?
6. In the **equal space environment**, what are the initial positions of the three agents?
7. In Table 2, how were the trajectories categorized into “good,” “normal,” and “poor”? Could the authors shed more light on how different qualities of datasets were collected?
8. MADT has been cited with citation number 18 in many places (including the tables), but citation 18 refers to MA-BCQ. The correct citation number for MADT is 27. The authors should thoroughly cross-check the citations in the main text and the supplementary.

**Limitations:**

- Authors haven't discussed potential negative societal impacts of their work.
- Authors have addressed the scalability of their method as a limitation.

**Strengths And Weaknesses:**

# Strengths

1. Leverages transformers for learning beneficial cross-agent interactions and distills them into student policies via well-motivated structural-relation loss constraint
2. Achieves strong empirical performance in various multi-agent environments
3. Authors conducted extensive ablation studies to show the importance of every component in the proposed method, as well as discussed.

# Weaknesses

1. Figure 6 compares only the proposed method w/ and w/o pretraining. Authors should include a comparison with MADT-online too.
2. My main concern is that the method does not seem scalable. In environments like SMAC (where number of agents > 2), simpler transformer-based method MADT performs comparable to the proposed method.
3. Minor: Lots of typos and grammatical errors in the text

---

> ### Author Response · Authors · 2022-07-31
> **Response to Reviewer Vg5F**
>
> We thank reviewer 3 for the detailed comments and helpful suggestions. The typo and citation number has been fixed in our revised version.
>
> Comment: **In the Algorithm, shouldn't the student-agent gradient update equation (φi = φi − α ∇φi (Li action + αLi rel + βLi KL)) come inside the inner for-loop?**
>
> - Response: Yes and thanks for pointing it out! We have fixed it in the revised paper.
>
> Comment: **Figure 6 compares only the proposed method w/ and w/o pretraining. Authors should include a comparison with MADT-online too.**
>
> - Response: Thanks for the suggestion. We have added the comparison in Figure 6 in the revised paper. The results show that the proposed method is more data efficient in online finetuning compared to MADT across all environments.
>
> Comment: **I am not sure how the Fill-In environment works. If agent 1 has passed a particular block, will that block be then inaccessible for agent 2, and vice-versa?**
>
> - Response: The goal of the Fill-In environment is to have agents cooperatively traverse all the blocks. At each timestep, the agent can choose to change direction or move forward. The reward is “−1 * the number of not traversed block + the number of the traversed block”. The first term of the reward informs agents how many blocks aren’t traversed yet, while the second term encourages agents to traverse novel blocks. Note that once a block has been traversed by an agent, other agents will not be rewarded to go through the same block. While we do not explicitly make passed block inaccessible, the setting with irreclaimable reward for each block still requires cross-agent credit assignment to achieve optimal strategy for maximal coverage. In our visualization of the environment (e.g., Fig. 1), we used colored triangle to represent each agent’s location and colored blocks to represent each agent’s trajectory. We have added a detailed description of the environment in the revised version.
>
> Comment: **Re: Table 11 - Why does the return distribution for the Highway dataset have the opposite order, i.e., poor > normal > good? It seems like a typo to me.**
>
> - Response: Yes, we’ve fixed it in the revised version.
>
> Comment: **Referring to Line 563 - why are obstacles and apples needed in the Fill-In environment? Is this a typo?**
>
> - Response: It is a typo and we have fixed it, thank you!
>
> Comment: **In the equal space environment, what are the initial positions of the three agents?**
>
> - Response: The initial positions of agents are randomly sampled from a uniform distribution.
>
> Comment: **How are the trajectories with different quality collected? In Table 2, how were the trajectories categorized into “good,” “normal,” and “poor”? Could the authors shed more light on how different qualities of datasets were collected?**
>
> - Response: In Grid-World and Highway, the “good” trajectories are collected with an expert policy that is trained to achieve saturated performance. The “normal” trajectories are collected with a policy that reaches around 80% performance of the expert policy, and the “poor” trajectories are collected with the policy that reaches around 70% performance of the expert policy. We have added these details in the revised supplementary material. Readers can find this information at table 11 of the supplementary material.
>
>   The detailed performances of the trajectories are summarized below:
>   | Dataset Quality                  | good             | normal           | poor             |
>   |----------------------------------|------------------|------------------|------------------|
>   | Grid-World - return distribution | $3.02 \pm 0.29$  | $2.50 \pm 0.12$  | $2.13 \pm 0.09$  |
>   | Highway - return distribution    | $26.89 \pm 1.92$ | $20.91 \pm 1.39$ | $18.71 \pm 1.53$ |
>
> Comment: **MADT has been cited with citation number 18 in many places (including the tables), but citation 18 refers to MA-BCQ. The correct citation number for MADT is 27. The authors should thoroughly cross-check the citations in the main text and the supplementary.**
> - Response: We sincerely apologize for the error and any potential negative impact our obliviousness has caused. We have fixed the citation and want to thank reviewer for pointing it out.

---

> > ### Comment · Reviewer_Vg5F · 2022-08-08
> > **Response**
> >
> > I thank the authors for addressing all of my queries. After reading the rebuttal from the authors, I would like to raise my score to a weak accept, as I think that the proposed method is well-motivated with strong empirical gains.

---

### Official Review · Reviewer_2W5g · 2022-07-12

**Rating:** 6
**Confidence:** 5
**Soundness:** 3 good
**Presentation:** 4 excellent
**Contribution:** 3 good

**Summary:**

In this paper, the authors propose a distillation-based offline MARL method for improving the quality of pre-trained multi-agent policy online. The authors construct the teach and student policy with access to the global and local information, repsectively. Experimental results on the intuitive and public environments show improvement over many strong baseline methods.

**Questions:**

1. The idea of this paper is similar to a recent work named Jump-Start Reinforcement Learning, which learn to use the guide policy to inform the whole online policy. Can you show the difference between the proposed approach of this method?

2. Can you show more detailed distillation process used in this work? And how to distilled a reasonable student policy?

**Limitations:**

The scalability may limit the application of this work, which the complexity will increase with the scale of multiple agent enlarges as described above.

**Strengths And Weaknesses:**

Strengths: The authors propose the utilization of global and local information with teach and student policy distillation.
1. The authors show the reasonable knowledge distillation on the teach policy pre-trained with global information. Then the student policy can be learned by distilled the parameter individually with relational constraints to improve the cooperation. The idea can be validated from the centralized training and decentralized execution (CTDE) framework.
2. The clear writing and sufficient experiments show the effectiveness of this approach compared with several strong baselines including sequential modelling on MARL, MADT.

Weakenesses: The learning complexity may increase with the scale of multi-agent enlarges.
1. I think this work has the obvious limitation in the computation complexity including learning the teacher policy from offline dataset, and constraining the relational distillation.
2. In line 296, the authors conclude the supereme in the poor offline data compared with MADT. I want to know the theoretical analysis of this results in detail.

---

> ### Author Response · Authors · 2022-07-31
> **Response to Reviewer 2W5g**
>
> Thank you for the helpful comments. Here are our replies.
>
> Comment: **The idea of this paper is similar to a recent work named Jump-Start Reinforcement Learning, which learn to use the guide policy to inform the whole online policy. Can you show the difference between the proposed approach of this method?**
>
> - Response: Thanks for the pointer. JSRL uses offline data to learn a “guide policy” that moves the agent to certain states that are closer to the goal states. Then, the “exploration” policy is deployed to improve itself through reinforcement learning (RL). The algorithm is designed to tackle tasks with *hard exploration challenges*. In contrast, our goal is to use a “teacher” policy (represented as a decision transformer) to bypass bootstrapping and perform credit assignment directly via self-attention. We focus on the offline setting and proposed to distill the knowledge of the teacher policy into student policies while JSRL focuses on the online setting by leveraging the "guide policy" to ease the exploration. In our revised manuscript, we have edited the Related Work section to discuss the difference between our method and JSRL.
>
> Comment: **The authors conclude the supereme in the poor offline data compared with MADT. I want to know the theoretical analysis of this results in detail.**
>
> - Response: The conclusion comes from empirical observation and we apologize for any implication that causes the reviewer to believe that there is theoretical analysis. We are happy to revise our paper to avoid confusion.
>
> Comment: **Can you show a more detailed distillation process used in this work? And how to distilled a reasonable student policy?**
>
> - Response: The first step is to train the teacher policy that is represented as a decision transformer. It takes observations, actions, and rewards from all agents (combined via concatenation) as inputs and predict all agents' actions for the next timestep.
>
>   The distillation process includes two procedures.
>   The first part is conventional knowledge distillation that minimizes $L_{KL}^i$, the KL divergence between the actions predicted by the teacher policy $\hat{a}^i$ and the student's policy $\overline{a}^i$. The second part is structural distillation, which makes the agent mimic the way that agents cooperate with each other. Let $\hat{f}^i$ and $\overline{f}^i$ be the teacher's features and the student's features in the decision transformer of the $i$'th agent, respectively. The structural distillation can be represented as: $L_{rel}^i = \sum_{j\neq i}\mathbf{H}(cos^{-1}(M(\hat{f}^i), M(\hat{f}^j)), cos^{-1}(N(\overline{f}^i), N(\overline{f}^j)))$. $\mathbf{H}$ denotes the Huber loss. $M$ and $N$ are mapping networks that are used to transform features. The intuition of our objective is to preserve the angles of teacher’s (transformed) features between agents during distillation so that students' (transformed) features have the similar angles. We refer the reviewer to Section 3.2 for more details on motivations of each component and design choice and we are happy to revise the section for better lucidity. Also, we identify several key factors of effective distillation, including mapping network, non-shared weights, structural distillation, etc. Empirical analysis and related discussion are shown in Table 3, Figure 5, and Section 4.2.

---

### Official Review · Reviewer_dysR · 2022-07-17

**Rating:** 5
**Confidence:** 4
**Soundness:** 3 good
**Presentation:** 2 fair
**Contribution:** 3 good

**Summary:**

The paper designs an offline MARL framework based on decision transformer and policy distillation. And proposes a novel distillation loss to distillate the structural relations among different agents from a centralized teacher policy. The experiment results illustrated the improvement brought by policy distillation.

**Questions:**

See weeknesses before.

**Limitations:**

The authors mentioned the scalability problem. Also, the tested environments most have relatively short horizons. I wonder whether this framework can still remains a relatively fast training speed (additional 10% training time) if the horizons are long since attention has O(L^2) complexity.

**Strengths And Weaknesses:**

Strength:
1. The idea of combine knowledge/policy distillation and decision transformer for multi-agent scenario is interesting.
2. The proposed design of relational policy distillation loss is reasonable in multi-agent cases.
3. The authors conduct thorough experiments on multiple environments. The proposed framework achieves state-of-the-art results and the ablation shows the performance raise brought by relational policy distillation loss is important.

Weaknesses:
1. Although the proposed framework achieves significant performance raise on the four environments designed by the authors. The improvements on SMAC benchmarks are not significant compared to MADT. In the ablation studies, the conventional distillation method performs worse than IDT or MADT on multiple tasks. Is this because the teacher policy itself does not take a big lead on these tasks?
2. The authors did not mention the settings for initial target return, which would significantly influence the performance of DT in single-agent settings. The setting of target return in multi-agent cases should be more complicated as it involves credit assignment.
3. The authors did not analysis the influence of number of parameters, which might be important as DT is more complicated than networks used in previous offline MARL methods.

---

> ### Author Response · Authors · 2022-07-31
> **Response to Reviewer dysR**
>
> Dear reviewer, thank you for the insightful questions!
>
> Comment: **In the ablation studies, the conventional distillation method performs worse than IDT or MADT on multiple tasks. Is this because the teacher policy itself does not take a big lead on these tasks?**
>
> - Response: We thank the reviewer for noticing this! We found that conventional distillation may force the student agents to “memorize” the teacher’s behavior while ignoring the relations between agents (see Fig. 5 and the paragraph starting at line 269 for an example). This causes student agents to misunderstand the task and thus results in worse performance. These results highlight the importance of the proposed structural distillation which can potentially represent how agents interact with each other. The teacher policy can achieve at least 5 percent improvement over baselines consistently across all tasks hence we believe that the teacher policy is not the bottleneck.
>
> Comment: **The authors did not mention the settings for initial target return, which would significantly influence the performance of DT in single-agent settings. The setting of target return in multi-agent cases should be more complicated as it involves credit assignment.**
>
> - Response: Thanks for asking this important question! The target return during training is the return of each agent within the offline trajectory. After executing the generated action for the current state, we decrement the target return by the achieved reward and repeat until episode termination. As for the evaluation phase, we choose return targets based on expert performance for each environment, which is similar to the strategy of choosing target return in the original decision transformer paper.
>
> Comment: **The authors did not analysis the influence of number of parameters, which might be important as DT is more complicated than networks used in previous offline MARL methods.**
>
> - Response: The number of parameters in a decision transformer is around 1.8M. To analyze the influence of the number of parameters, we scale up the model architecture of MA-ICQ and MA-CQL to around 1.8M parameters as well and re-run the same experiments.
>   Here are the results:
> | method  | Ours            | MA-ICQ         | MA-CQL          |
> |---------|-----------------|----------------|-----------------|
> | Fill-In | $-3.41\pm0.12$ | $-9.98\pm1.97$ | $-10.21\pm0.81$ |
>
> We find that our method still outperforms the baselines even after scaling them up to match the number of parameters of our method.

---

### Meta-Review · Area_Chair_C58r · 2022-08-25

**Recommendation:** Accept
**Confidence:** Less certain

**Metareview:**

This paper deals with offline multi-agent RL tasks. Based on decision transformer, the authors propose to first train a centralized decision transformer over the offline data to capture the agent interaction patterns and then distill the knowledge from the centralized decision transformer to the individual agents. The idea is clear and the experiment looks comprehensive. As mentioned by the reviewers, the performance of the proposed method looks just comparable to MADT, which is a very simple parameter-sharing architecture.


**Award:**

No

---

### Decision · Program_Chairs · 2022-09-14

Accept